# Effects of Non-Essential Amino Acids on Knee Joint Conditions in Adults: A Randomised, Double-Blind, Placebo-Controlled Trial

**DOI:** 10.3390/nu14173628

**Published:** 2022-09-02

**Authors:** Fumika Takeuchi, Michihiro Takada, Yasuo Kobuna, Hirohisa Uchida, Yusuke Adachi

**Affiliations:** 1Research Institute for Bioscience Products & Fine Chemicals, Ajinomoto Co., Inc., Kawasaki 210-8681, Kanagawa, Japan; 2Kobuna Orthopedic Clinic, Maebashi 371-081, Gunma, Japan; 3Sports Nutrition Department, Ajinomoto Co., Inc., Chuo-ku, Tokyo 210-8681, Japan; 4Institute of Food Sciences and Technologies, Ajinomoto Co., Inc., Kawasaki 210-8681, Kanagawa, Japan

**Keywords:** amino acids, joint, VAS, JKOM, JOA

## Abstract

Joint problems impair performance during exercise and daily activities and influence quality of life. The present study aimed to examine the effects of a combination of six non-essential amino acids (6AA) on joint conditions in an adult population. A total of 50 participants aged between 20 and 64 years with joint discomfort but no diagnosed joint disorder were randomly and blindly assigned to a control or 6AA group. The 6AA group took 12 g of the non-essential amino acid formulation orally (4 g three times a day) and the control group took equivalent doses of a placebo. Each group maintained the daily dose for 12 weeks. Primary outcome measures were evaluated with the visual analogue scale (VAS), the Japanese Knee Osteoarthritis Measure (JKOM), and the Japanese Orthopaedic Association score (JOA). These tests were taken before the experiment began at 4 weeks and 12 weeks after the intervention. The results of the VAS indicated that 6AA improved joint pain, discomfort, and stiffness both during a resting state and during normal activity. Participants’ scores on the JKOM and JOA also showed significant improvements in the group that had taken the 6AA supplement. These results demonstrate that 6AA improves symptoms of joint problems, such as pain, discomfort, stiffness, and difficulty in performing daily activities after 4 weeks of daily consumption.

## 1. Introduction

Joints are composed of bone, articular cartilage, joint capsules, synovium, and tendons. They play a role in locomotion and mitigate the physical impact. Joint problems impair movement during exercise and daily activities, resulting in reduced quality of life (QOL). With age, joint cartilage wears out as it loses its elasticity, leading to conditions such as arthritis, which may shorten a healthy life span [1,2].

Joint tissue is primarily composed of the extracellular matrix (ECM) produced by cells. This contains chondroitin, keratosulfate, and various types of collagens. Recently, it has been reported that collagen and gelatine in food improve joint discomfort and functionality [3,4,5]. Collagen in human joint tissue is a protein constructed mainly from non-essential amino acids [6,7]. Consumption of collagen increases plasma levels of the non-essential amino acids, glycine and proline [8]. These non-essential amino acids have been reported to improve arthritis and promote collagen synthesis in the cells of animals [9,10]. Thus, the non-essential amino acids abundant in collagen appear to play an important role in maintaining joint tissue.

This study aimed to determine the effects of non-essential amino acids on joint conditions in humans. A supplement containing six non-essential amino acids (alanine, aspartic acid, glutamic acid, glycine, proline, and serine) abundant in collagens [6,7] was used in this study. The supplement was taken three times a day for 12 weeks by otherwise healthy individuals experiencing joint discomfort. The condition of participants’ knee joints was assessed using the Japanese Knee Osteoarthritis Measure (JKOM) and Japanese Orthopaedic Association (JOA) scores, which are both clinically authoritative evaluation indices. In addition, joint discomfort, pain, and stiffness were evaluated using the visual analogue scale (VAS).

## 2. Materials and Methods

### 2.1. Study Design and Participants

This study was a randomised, double-blind, placebo-controlled, parallel-group 12-week trial, followed by a 12-week posttreatment observation period. Participants aged 20–64 years who met the study criteria were recruited from those on the clinical trial register of MC-Connect Corporation (Gunma, Japan). An invitation e-mail was sent to suitable potential participants, containing the conditions for participation. Those aware of discomfort in one or both knee joints were asked to participate in a screening test. Screening tests were conducted with 150 individuals in the two weeks preceding the start of the study by the principal investigator. Screening consisted of measurement of VAS, JKOM and JOA scores; knee X-rays to determine Kellgren–Lawrence (K–L) grades [11]; and blood and urine collection for biochemical analyses and collection of general demographic information.

The inclusion criteria were as follows: the participants were: (1) Aged between 20–64 years; (2) Graded 0 or I on the K-L test by the principal investigator; (3) Living normally but with discomfort in one or both knee joints; (4) Capable of independent decision-making and the provision of written informed consent. We excluded individuals who: (1) Had a history of rheumatoid arthritis or could be suffering from rheumatoid arthritis based on a blood marker test (rheumatoid factor > 15 mg/dL); (2) Were diagnosed with osteoarthritis requiring treatment; (3) Had an implanted artificial joint; (4) Were on medications that might affect the knee joint, such as poultices, ointments or painkillers; (5) Had joint dysfunctions, such as meniscus injury or were receiving treatment for such a dysfunction; (6) Had previously undergone surgery for chronic knee pain; (7) Were taking supplements or eating functional foods that may affect the knee joint during the study and were unable or unwilling to stop during the study period; (8) Were taking amino acid-related medicines, such as Aminoleban^®^; (9) Were regularly taking amino acid or protein supplements; (10) Had an amino acid metabolism disorder (confirmed whether the participants had congenial amino acid metabolic disorders by self-report); (11) Were pregnant, lactating or trying to conceive; (12) Had undergone blood tests or given blood, with collection of more than 200 mL of blood, within 12 weeks of the screening test; (13) Had participated in a study requiring or restricting the intake of medicines or specific foods, or involving the use of cosmetics or skin preparations within the preceding month, or would be participating in other research or receiving medical intervention during the study period; (14) Were judged as unsuitable for the study by the principal investigator for other reasons.

The sample size was calculated based on a previous study [12] in which the JOA score changed significantly after 12 weeks of intervention. The mean effect size and standard deviation (SD) for our study were estimated as 10.3 and 10.3, respectively, based on the results of the previous study. With a two-sided 5% significance level, the sample size required to achieve 90% power was 44 (22 per group) based on the mean effect size and standard deviation described above. Sample size calculation was based on two sample *t*-tests. We used the R function power.t.test via R version 3.4.1 (https://cran.r-project.org (accessed on 26 April 2019)). Allowing for dropouts, the final sample size was determined to be 25 participants per group (50 participants in total).

The allocation manager randomly allocated the 50 participants to either the six amino acids (6AA) supplementation (experimental) group or the placebo supplementation (control) group by a stratified block randomisation. The random allocation was then adjusted to ensure equality between-group distribution of sex, age, and baseline JKOM and JOA scores. The link between identification number of the participants and assigned treatment group was kept in a sealed document by the allocation manager. The allocation manager was not involved in assessments of eligibility, data collection, or analysis. The group allocation of each participant was concealed from the researchers, clinicians, and participants until the final analyses were complete. The allocation table was sealed and stored until the time of key opening by an independent controller. This study was performed by a contract research organisation, KSO Corporation (Tokyo, Japan), from April to December 2020 at Kobuna Orthopedic Surgical Clinic (Maebashi, Japan).

### 2.2. Interventions

The supplement used in this study was a granular powder containing six non-essential amino acids, namely, DL-alanine (1.16 g), sodium L-aspartate (0.56 g), L-glutamic acid (0.51 g), glycine (0.56 g), L-serine (0.51 g), and L-proline (0.68 g). The placebo was prepared by replacing the amino acids with maltitol. The test product was packed in single aluminium sachets, each containing 4 g. The participants ingested one sachet of the supplement or placebo with approximately 200 mL of water three times a day for 12 weeks. To eliminate any effects of combining the supplement with food, participants were instructed to ingest it ≥2 h after breakfast and dinner and before bed. The level of compliance was calculated by collecting all opened and unopened sachets and remaining supplements.

### 2.3. Outcome Measurement

Baseline measures were taken of subjective symptoms and blood urine parameters at week 0 (W0), before the intervention; week 4 (W4); and week 12 (W12), after the intervention. The primary outcome was the condition of the participants’ knee joints. This was measured using the VAS, JKOM, and JOA scales.

The VAS is a reliable, validated self-report measure of the subjective experience of pain [13]. In this instance, it was used to determine the intensity of pain, discomfort, and stiffness in the knee. The VAS scores are standardised, and participants mark a point on a 100 mm horizontal line to represent their pain, discomfort, or stiffness level. One end of the line is marked 0 and represents no pain/discomfort/stiffness; the other end is marked 100 and represents the worst possible pain/discomfort/stiffness. The distance between 0 and the point marked by the participant is measured in millimetres and recorded as the VAS score. The VAS scores were obtained for the pain, discomfort, or stiffness experienced during eight different activities. These were as follows: VAS1, walking upstairs; VAS2, walking downstairs; VAS3, standing up from a chair; VAS4, kneeling (‘seiza’ in Japanese); VAS5, standing from seiza; VAS6, getting out of bed upon waking; VAS7, at rest before bed; and VAS8, walking a long distance or for a longer time than usual.

The JKOM is a well-established assessment tool for the measurement of pain, discomfort, and stiffness in the knee [14]. The JKOM-I assessed overall knee pain (one question) along with another four domains comprising 25 questions, making the total number of questions 26. The four additional domains included eight questions on knee pain and stiffness over the last few days (JKOM-II), ten questions regarding problems in daily life due to knee pain over the last few days (JKOM-III), five questions regarding the effect of knee pain on their usual activities in the last month (JKOM-IV), and two questions regarding their general health over the last month (JKOM-V). Participants provided answers to each question on a 5-point Likert scale from no impairment (0 points) to serious impairment (4 points). The responses were summed to produce a total score ranging from 0 to 100 points. The total scores and the scores for each domain were compared between the experimental and control groups.

The JOA is used to evaluate knee conditions from the clinician’s point of view [15]. The JOA scores several aspects of the condition of the knee with higher scores indicating less joint impairment. Scores are collected for pain when walking (0–30), pain when ascending or descending stairs (0–25), range of motion (0–35) and joint effusion (swelling; 0–10). This allows a total score range of 0–100.

The secondary outcomes for this study were serum levels of the cartilage type II collagen degradation products, C-terminal telopeptide of type II collagen (CTX-II) and type II collagen cleavage neoepitope (C2C), and uric acid and plasma levels of the inflammatory marker and tumour necrosis factor-alpha (TNF-α). The two collagen products are useful markers of cartilage status [16,17]. Serum uric acid was measured because the association between blood uric acid levels and knee conditions have previously been reported [18,19,20]. The TNF-α is a biomarker of inflammatory processes and contributes to the destruction of cartilage [21]. Therefore, higher levels of plasma TNF-α can be used to indicate poorer joint condition.

### 2.4. Safety Evaluation

To assess the safety of the 6AA supplement, measurements were taken of body mass index (BMI), blood pressure, and pulse rate, and blood and urine tests were conducted at W0, W4, and W12. The blood and urine samples were collected after overnight fasting. The following blood parameters were measured: white blood cell count; red blood cell count; haemoglobin (Hb); haematocrit (Ht); platelet count; total protein; albumin; total bilirubin; aspartate aminotransferase (AST); alanine aminotransferase (ALT); lactate dehydrogenase (LDH); alkaline phosphatase (ALP); γ-glutamyl transpeptidase (γGTP); blood urea nitrogen; creatinine; sodium (Na); chlorine (Cl); potassium (K); total cholesterol; LDL-cholesterol; HDL-cholesterol; triglyceride (TG); fasting plasma glucose concentration; and glycated haemoglobin (HbA1c). The urine parameters measured were protein, urinary glucose, and urinary occult blood.

### 2.5. Statistical Analysis

Data were described as the means and SDs, or 95% confidence interval (CI). Efficacy and safety analyses were conducted on the full analysis set. The statistical significance of between-group differences was assessed by Student’s *t*-test, comparing changes from baseline W0 to W4 and from W0 to W12. To analyse the time course of effects, linear mixed models (LMMs) [22] were applied with random intercepts. The LMMs were modelled using the group, the number of weeks (time), and their interaction as fixed effects. The significance of between-group differences was investigated according to the interactions between the groups and times. A two-sided *p*-value of 0.05 was considered statistically significant. All statistical analyses were performed using R version 4.0.3 (R Foundation for Statistical Computing, Vienna, Austria). The statistical analysis plans were finalized before the key opening.

## 3. Results

### 3.1. Participants and Compliance

Of the 150 participants screened, 100 were excluded (Figure 1). This left a total of 50 participants who were randomly and equally allocated to either the 6AA or placebo group. The mean 6AA intake rate of participants was 99.98%. There were no dropouts and all participants complied with the study requirements and completed the trial. Thus, the dataset analysed included the data of all study participants.

The clinical and demographic baseline data of the participants are summarised in Table 1. There were no significant differences in age, sex, height, body weight, BMI, body fat, or skeletal muscle index (SMI) between the groups. All participants had a K–L grade of 0 or I. The number of participants with each of these K–L grades in the experimental and control groups was almost equal.

### 3.2. Effects of the Intervention

In this study, the primary outcome was the joint condition. This was evaluated by VAS, JKOM, and JOA. There were no significant between-group differences in the scores for these three measures at W0. The VAS scores for pain, discomfort, and stiffness in the knee joint are shown in Table 2. The 6AA group showed a significantly greater reduction than the control group in all VAS scores from W0 to W4 and W12. This difference was significant when analysed by both the unpaired *t*-test and LMM (*p* < 0.05).

Table 3 shows the results of the JKOM evaluation of knee conditions for the two groups at the three measurement time points. The 6AA group had significantly lower scores than the control group on the JKOM at both W4 and W12. The LMM analysis found a significant difference between the total JKOM scores of each group. Comparisons between the experimental and control group in each JKOM domain (I–V) were made using unpaired *t*-tests. Significant between-group differences were detected in domains I, II, and III at both W4 and W12. In domain V, the 6AA group showed a significantly greater improvement in scores than the control group at the end of the intervention period (*p* = 0.0052). No significant differences between the two groups were observed in domain IV. The LMM analysis confirmed that the 6AA group score reductions were significantly greater than those of the control group in domains I, II, III and V (*p* < 0.05). There was a greater reduction in the JOA scores of the 6AA group than the control group at W12, but this was non-significant (*p* = 0.0901). However, the LMM analysis found significantly greater reductions in JOA scores for the 6AA group than for the control group (*p* = 0.021).

Serum CTX-II, C2C, uric acid, and plasma TNF-α levels (from samples taken at the three measurement time points) were secondary outcome measures for this study. Only uric acid was significantly different between the 6AA and placebo groups (Table 4).

### 3.3. Safety Evaluations

There were no abnormalities in the blood or urine laboratory test results or clinical data of our participants that could lead to safety concerns. Further assessment of supplement safety was made through medical interviews with each participant and no issues were reported. Appendix A shows the results of the blood and urine laboratory tests.

## 4. Discussion

The present study aimed to evaluate the effects of 6AA intake on knee joint discomfort in subjects. The primary outcome was measured by comparison of the changes in VAS, JKOM, and JOA scores after 4 and 12 weeks of participants in the 6AA group and the placebo group. For VAS scores, this comparison of changes in VAS scores demonstrated that pain, discomfort, and stiffness in knee joints are significantly improved by the intake of 6AA (Table 2). The JKOM is a questionnaire for assessing the extent of knee problems. Similarly, supplementation with 6AA produced significantly greater improvement in total JKOM scores than supplementation with the placebo. Consistent with the measurement of VAS scores, JKOM-I and II scores are related to pain, discomfort, and stiffness, and consistent with the VAS results, these scores were significantly improved by 6AA supplementation. Also, significant decreases in JKOM-III and JKOM-V scores were observed in the 6AA group. This is because these domains are concerned with the effects of the knee condition on daily life and with subjective health status, and this result suggests that the QOL of participants was improved by the 6AA supplement. The LMM analysis of JOA scores showed significantly greater improvements in the 6AA group than in the placebo group. These results demonstrate that 6AA supplementation mitigates pain, discomfort, and stiffness in the knees both in a sedentary state and during daily activities, resulting in improved QOL.

Collagen accounts for about two-thirds of the dry weight of adult articular cartilage [23]. It is the most important structural and functional component of the ECM and is vital to the strength, regulation, and regeneration of articular cartilage [24]. Several studies have found that supplementation with collagen and related compounds, such as collagen hydrolysates, has beneficial effects on joint pain and discomfort. Although the mechanism responsible for the effects of collagen consumption on joint conditions is not fully established, possible explanations have been identified. The intake of hydrolysed collagen is known to stimulate chondrocytes to increase ECM synthesis, and this includes the synthesis of collagen in articular cartilage [25]. A cellular study found that type II collagen promotes the release of anti-inflammatory cytokines, such as interleukin-10, in immune cells and chondrocytes and facilitates ECM replenishment by chondrocytes [26]. Therefore, consumed collagen may enhance, repair, or regenerate deteriorating ECM collagen by stimulating ECM synthesis and suppressing inflammation in chondrocytes.

Orally-administered collagen is absorbed from the intestine during digestion and enters the bloodstream as peptides and free amino acids. These can be delivered to synovial joints via capillaries, directly affecting ECM metabolism in articular cartilage and immune cells such as macrophages [27]. As described above, there have been several studies showing the direct effects of collagen derivatives on cultured chondrocytes and immune cells [26,27,28,29]. In contrast, there have been relatively few studies on the mechanism behind the effects of free amino acids on joint health. Comparatively, there have been many more studies on the effects of glycine, which is most abundant in collagen [9,30]. Therefore, it has been hypothesised that glycine makes the greatest contribution to the beneficial effects of collagen supplements on joints. Like collagen peptides, glycine has also been reported to increase the synthesis of innate collagen and appears to suppress inflammation in cultured cell experiments [9,31]. However, whether a combination of amino acids that includes glycine affects joint conditions in humans has not been examined. In the present study, we tested the effects of a combination of six free amino acids on the human knee joint. Although we did not isolate the individual contributions of each of the non-essential amino acids in the 6AA supplement, our study provided new insight into the role of free amino acids in the maintenance and improvement of joint health.

This study had some limitations. Firstly, to evaluate the effects of 6AA on joint conditions, questionnaires were used as our primary outcome measures. While the measures used have proven reliability and validity [32] and the results were consistent between them, self-report measures of medical conditions cannot be regarded as objective and stringent evidence. To estimate the type II collagen degradation in cartilage, serum CTX-II and C2C were measured. The data showed no significant differences in those markers between the placebo and 6AA groups. Many previous studies have used urinary CTX-II and C2C levels as a measure of collagen degradation in cartilage [16,17,33,34], yet it is unclear whether serum CTX-II and C2C are equally appropriate for the evaluation of collagen degradation. On the other hand, the concentration of uric acid in the 6AA group was lower than that of the placebo group (Table 4). Since some studies indicate a relationship between serum uric acid levels and joint conditions, the reduced uric acid level may reflect improved joint conditions as a result of the 6AA supplementation. To examine the effects of 6AA in future studies, more appropriate markers should be selected. Secondly, dietary records were not obtained in this study. Thus, nutrients in the diets of our participants, particularly the non-essential amino acids found in foods, may have affected our outcomes. In future studies, dietary intake of nutrients should be assessed. Thirdly, only one dose was tested. All participants ingested 12 g/day (4 g three times per day). The effective dose of 6AA may be lower than this and further research is required to determine the optimum dose.

In conclusion, this trial demonstrated that supplementation with 6AA at doses of 12 g/day improved knee joint health and reduced symptoms of pain, discomfort, and stiffness in otherwise healthy individuals experiencing joint discomfort. The compliance rate was almost 100% throughout the study, and no adverse events were observed. To our knowledge, this is the first study to report the beneficial effects of free amino acids on the knee joint. Further study is required to clarify the mechanism underlying these effects and to determine the individual influences of each of the amino acids.

## Figures and Tables

**Figure 1 nutrients-14-03628-f001:**
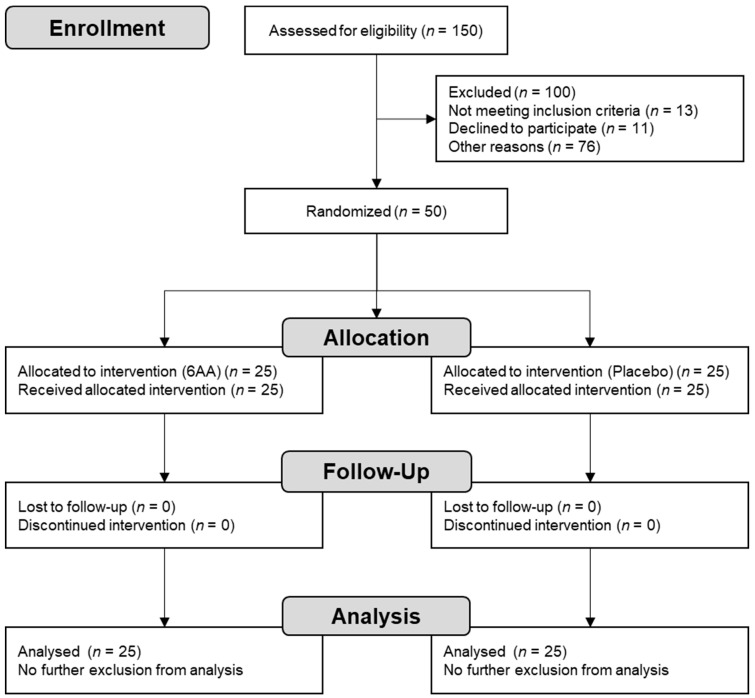
Flow chart of subject recruitment, randomisation, and follow-up.

**Table 1 nutrients-14-03628-t001:** Baseline characteristics of participants.

		6AA (*n* = 25)	Placebo (*n* = 25)	Difference (95% CI)	*p*-Value
Age (years)		49.8 ± 7.0	50.0 ± 5.1	−0.1 (−3.4 to 3.6)	0.9452
Sex (female/male)		22/3	21/4	-	1.0000
Height (cm)		158.6 ± 6.8	159.9 ± 6.3	−1.3 (−2.4 to 5.0)	0.4923
Body weight (kg)		56.3 ± 9.2	58.0 ± 9.6	−1.7 (−3.7 to 7.0)	0.5339
BMI (kg/m^2^)		22.3 ± 2.5	22.6 ± 2.8	−0.3 (−1.2 to 1.8)	0.6751
Body fat (%)		30.2 ± 6.1	30.0 ± 6.1	0.2 (−3.7 to 3.3)	0.9064
SMI (kg/m^2^)		21.1 ± 3.4	22.0 ± 4.4	−0.9 (−1.3 to 3.1)	0.4078
K–L grade	0	14	15	-	1.0000
	I	11	10	-	1.0000

Data are represented as mean ± SD; CI, confidence interval; significance between groups was assessed by Student’s *t*-test for continuous variables and Fisher’s exact test for categorical variables.

**Table 2 nutrients-14-03628-t002:** Visual analogue scale (VAS) scores at W0, W4, and W12.

		W0		W4				W12				*p*-Value LMM
		Values		Value		Difference (95% CI)	*p*-Value ^1^	Values		Difference (95% CI)	*p*-Value ^1^	
		6AA	Placebo	6AA	Placebo			6AA	Placebo			
VAS1	Pain	38.0	33.1	30.0	32.7	−7.6	0.0001 *	19.7	26.9	−12.2	0.0123 *	0.0023 *
		(22.2)	(20.2)	(20.6)	(20.7)	(−11.1 to −4.2)		(20.1)	(17.6)	(−21.6 to −2.8)		
	Discomfort	38.7	34.8	28.6	34.4	−9.7	0.0001 *	19.3	27.6	−12.2	0.0121 *	0.0040 *
		(22.7)	(20.9)	(21.0)	(21.7)	(−14.3 to −5.0)		(18.9)	(18.2)	(−21.5 to −2.8)		
	Stiffness	35.5	30.6	27.8	29.8	−7.0	0.0023 *	19.6	25.9	−11.3	0.0250 *	0.0055 *
		(24.8)	(20.2)	(21.6)	(19.7)	(−11.3 to −2.6)		(18.5)	(18.0)	(−21.2 to −1.5)		
VAS2	Pain	46.1	42.0	34.6	41.2	−10.8	0.0004 *	25.4	34.6	−13.4	0.0059 *	0.0024 *
		(22.4)	(22.4)	(23.3)	(22.0)	(−16.5 to −5.0)		(22.3)	(20.5)	(−22.7 to −4.0)		
	Discomfort	44.6	41.3	34.0	41.2	−10.4	0.0003 *	24.4	33.9	−12.8	0.0093 *	0.0041 *
		(24.3)	(22.1)	(24.1)	(22.1)	(−15.8 to −5.1)		(22.7)	(19.4)	(−22.2 to −3.3)		
	Stiffness	45.5	39.0	35.1	39.6	−11.0	0.0001 *	25.2	33.8	−15.1	0.0019 *	0.0006 *
		(24)	(21.7)	(24.7)	(21.6)	(−16.1 to −5.9)		(22.0)	(21.5)	(−24.4 to −5.9)		
VAS3	Pain	50.9	47.4	38.4	45.4	−10.4	0.0001 *	26.8	36.4	−13.2	0.0105 *	0.0046 *
		(21.4)	(21.5)	(22.1)	(21.5)	(−15.3 to −5.6)		(20.6)	(21.4)	(−23.1 to −3.2)		
	Discomfort	50.3	46.8	37.2	45.6	−11.9	<0.0001 *	26.2	36.3	−13.6	0.0072 *	0.0034 *
		(21.8)	(21.5)	(22.3)	(20.6)	(−16.9 to −6.9)		(18.7)	(21.8)	(−23.3 to −3.9)		
	Stiffness	48.7	42.4	36.5	42.0	−11.7	<0.0001 *	26.5	34.9	−14.7	0.0022 *	0.0006 *
		(22.0)	(22.3)	(22.2)	(21.4)	(−16.7 to −6.8)		(18.7)	(22.6)	(−23.9 to −5.6)		
VAS4	Pain	37.3	30.4	25.5	31.6	−13.0	<0.0001 *	19.2	26.4	−14.1	0.0025 *	0.0012 *
		(23.1)	(19.1)	(19.9)	(22.3)	(−18.1 to −7.9)		(18.5)	(16.8)	(−23.0 to −5.2)		
	Discomfort	39.4	30.9	26.8	32.0	−13.8	<0.0001 *	20.2	27.2	−15.6	0.0011 *	0.0003 *
		(22.8)	(22.7)	(19.9)	(24.1)	(−18.2 to −9.4)		(19.6)	(19.1)	(−24.6 to −6.5)		
	Stiffness	38.5	30.0	26.5	30.0	−12.0	<0.0001 *	19.9	26.0	−14.6	0.0037 *	0.0006 *
		(23.8)	(21.8)	(20.4)	(22.1)	(−16.4 to −7.6)		(19.2)	(19.0)	(−24.2 to −5.0)		
VAS5	Pain	40.5	39.3	30.4	41.2	−11.9	<0.0001 *	22.9	33.2	−11.5	0.0131 *	0.0083 *
		(21.5)	(20.6)	(21.5)	(20.9)	(−16.4 to −7.4)		(20.1)	(19.5)	(−20.4 to −2.5)		
	Discomfort	41.3	39.7	29.3	41.1	−13.3	<0.0001 *	22.1	33.6	−13.1	0.0074 *	0.0042 *
		(22.5)	(21.0)	(21.1)	(21.2)	(−18.5 to −8.2)		(19.1)	(20.2)	(−22.5 to −3.7)		
	Stiffness	39.1	37.8	28.4	39.7	−12.6	<0.0001 *	21.8	32.7	−12.1	0.0112 *	0.0059 *
		(21.8)	(22.3)	(21.5)	(21.0)	(−16.6 to −8.5)		(19.1)	(20.4)	(−21.4 to −2.9)		
VAS6	Pain	44.9	44.5	35.8	44.9	−9.5	<0.0001 *	26.5	36.3	−10.2	0.0228 *	0.0131 *
		(22.1)	(19.3)	(20.2)	(19.6)	(−13.1 to −5.8)		(20.9)	(19.9)	(−18.9 to −1.5)		
	Discomfort	45.0	44.4	34.6	44.8	−10.8	<0.0001 *	26.0	36.1	−10.8	0.0166 *	0.0101 *
		(22.5)	(19.9)	(20.3)	(20.2)	(−14.5 to −7.1)		(20.3)	(20.2)	(−19.5 to −2.0)		
	Stiffness	43.5	40.8	33.7	42.9	−11.9	<0.0001 *	26.2	34.8	−11.2	0.0128 *	0.0088 *
		(23.4)	(22.0)	(20.9)	(21.2)	(−15.8 to −8.0)		(20.5)	(21.0)	(−20.0 to −2.5)		
VAS7	Pain	23.8	17.3	15.7	19.1	−10.0	0.0002 *	11.0	17.4	−12.9	0.0005 *	0.0001 *
		(20.7)	(19.0)	(16.0)	(20.3)	(−14.9 to −5.0)		(12.5)	(17.4)	(−19.8 to −6.0)		
	Discomfort	24.1	16.2	15.8	19.5	−11.6	<0.0001 *	11.5	17.4	−13.8	0.0006 *	0.0001 *
		(20.9)	(19.6)	(16.4)	(21.1)	(−16.7 to −6.5)		(12.9)	(17.8)	(−21.3 to −6.2)		
	Stiffness	23.1	16.0	15.6	18.7	−10.2	<0.0001 *	11.6	17.3	−12.8	0.0008 *	0.0001 *
		(20.7)	(18.9)	(16.9)	(20.5)	(−14.7 to −5.8)		(12.8)	(17.7)	(−19.9 to −5.6)		
VAS8	Pain	45.1	36.3	31.1	35.2	−12.9	<0.0001 *	24.3	30.0	−14.6	0.0018 *	0.0010 *
		(20.8)	(21.0)	(17.9)	(20.1)	(−18.1 to −7.7)		(18.6)	(18.2)	(−23.4 to −5.7)		
	Discomfort	47.2	37.2	30.6	37.1	−16.4	<0.0001 *	22.2	31.1	−18.8	0.0001 *	<0.0001 *
		(20.5)	(21.5)	(17.3)	(19.6)	(−22.0 to −10.8)		(15.8)	(19.3)	(−27.8 to −9.9)		
	Stiffness	43.8	34.1	29.4	34.3	−14.6	<0.0001 *	21.7	29.3	−17.3	0.0001 *	<0.0001 *
		(20.8)	(21.4)	(17.0)	(20.1)	(−20.1 to −9.0)		(15.2)	(19.2)	(−25.7 to −9.0)		

Data are presented as mean ± SD. ^1^ CI, confidence interval; LMM, linear mixed model; SD, standard deviation; W, week. The differences between the 6AA and control groups in changes from W0 to W4 and W12 were assessed by Student’s *t*-test * *p* < 0.05 between the 6AA and placebo groups. VAS scores were obtained for pain, discomfort, or stiffness experienced during eight different activities. These were as follows: VAS1, walking up stairs; VAS2, walking down stairs; VAS3, standing up from a chair; VAS4, kneeling (‘seiza’ in Japanese); VAS5, standing from seiza; VAS6, getting out of bed upon waking; VAS7, at rest before bed; and VAS8, walking a long distance or for a longer time than usual.

**Table 3 nutrients-14-03628-t003:** Japanese Knee Osteoarthritis Measure (JKOM) and Japanese Orthopaedic Association (JOA) scores at W0, W4, and W12.

		W0		W4				W12				*p*-Value LMM
		Values		Values		Difference (95% CI)	*p*-Value ^1^	Values		Difference (95% CI)	*p*-Value ^1^	
		6AA	Placebo	6AA	Placebo			6AA	Placebo			
JKOM	I	45.0	37.7	36.9	35.4	−5.8	0.0199 *	23.1	31.8	−15.9	0.0004 *	<0.0001 *
		(18.4)	(16.3)	(19.2)	(15.0)	(−10.6 to −1.0)		(19.1)	(14.4)	(−24.4 to −7.5)		
	II	7.9	7.9	6.1	7.3	−1.2	0.0124 *	4.6	6.6	−2.0	0.0094 *	0.0026 *
		(3.1)	(2.4)	(3.6)	(2.5)	(−2.1 to −0.3)		(3.3)	(2.2)	(−3.6 to −0.5)		
	III	4.9	5.1	3.0	5.0	−1.9	0.0002 *	2.6	4.6	−1.8	0.0001 *	0.0010 *
		(4.0)	(3.3)	(4.0)	(3.3)	(−2.8 to −0.9)		(3.6)	(3.2)	(−2.6 to −1.0)		
	IV	3.3	4.0	2.7	3.8	−0.3	0.4826	2.6	3.4	0	0.9357	0.9533
		(2.3)	(2.6)	(2.0)	(2.2)	(−1.2 to 0.6)		(2.0)	(2.3)	(−1.0 to 1.0)		
	V	1.8	1.9	1.6	1.8	−0.1	0.3534	1.1	1.7	−0.5	0.0222 *	0.0052 *
		(0.9)	(0.9)	(0.9)	(0.9)	(−0.4 to 0.1)		(1.0)	(0.7)	(−1.0 to −0.1)		
	Total	17.9	18.9	13.4	17.9	−3.5	0.0014 *	10.9	16.3	−4.4	0.0016 *	0.0010 *
	(II–V)	(8.9)	(7.4)	(8.9)	(7.4)	(−5.6 to −1.4)		(8.4)	(7.0)	(−7.0 to −1.8)		
JOA	Mean	91.0	91.2	91.8	91.8	0.2	0.7305	93.6	92.1	1.7	0.0901	0.0210 *
		(3.0)	(2.5)	(3.4)	(2.6)	(−1 to 1.4)		(4.0)	(3.2)	(−0.3 to 3.7)		

Data are represented as mean (SD); CI, confidence interval; ^1^ Student’s *t*-test assessed the differences between the 6AA and placebo groups of change from W0 at W4 and W12 weeks; LMM, linear mixed model; * *p* < 0.05 between the 6AA and placebo groups.

**Table 4 nutrients-14-03628-t004:** The serum CTX-II, C2C, uric acid, and plasma TNF-α levels at W0, W4, and W12.

	W0		W4				W12				*p*-Value LMM
	Values		Values		Difference (95% CI)	*p*-Value ^1^	Values		Difference (95% CI)	*p*-Value ^1^	
	6AA	Placebo	6AA	Placebo			6AA	Placebo			
CTX-II	465.0	487.6	465.0	501.3	−13.7	0.4462	485.7	479.0	29.3	0.1817	0.0680
(pg/mL)	(167.2)	(192.2)	(165.5)	(181.4)	(−49.6 to 22.2)		(159.9)	(165.8)	(−14.2 to 72.7)		
C2C	330.4	339.2	346.4	358.2	−3.1	0.7825	364.5	354.2	19.1	0.2726	0.1551
(ng/mL)	(51.3)	(40.3)	(38.7)	(39.1)	(−25.7 to 19.5)		(70.5)	(43.3)	(−15.5 to 53.8)		
Uric acid	4.6	4.9	4.3	5.6	−0.9	0.0002 *	4.4	5.4	−0.5	0.0383 *	0.0407 *
(mg/dL)	(1.2)	(1.2)	(1.1)	(1.1)	(−1.3 to −0.4)		(1.1)	(1.2)	(−0.9 to 0)		
TNF-α	4.9	5.1	5.1	5.4	−0.1	0.6805	4.7	5.1	−0.2	0.6353	0.6288
(pg/mL)	(1.6)	(2.5)	(1.8)	(2.7)	(−0.8 to 0.6)		(2.1)	(3.2)	(−1.0 to 0.6)		

Data are represented as mean (SD); CI, confidence interval; LMM, linear mixed model; ^1^ Student’s *t*-test assessed the differences between the 6AA and placebo groups of change from W0 at W4 and W12 weeks. * *p* < 0.05 between the 6AA and placebo groups.

## Data Availability

Data sharing not applicable.

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
