# Peer review of "Effects of Non-Essential Amino Acids on Knee Joint Conditions in Adults: A Randomised, Double-Blind, Placebo-Controlled Trial"

_nutrients, 2022, doi:10.3390/nu14173628_

Round 1

Reviewer 1 Report

The authors investigated the role of non-essential amino acids in knee joint conditions in healthy adults. The study is interesting and brings a perspective on the use of these amino acids to help improve, mainly, pain, discomfort and stiffness. However, there are some gaps that could be filled to improve the manuscript.   2.1. Study design and participants. In the exclusion criterion "(10) had an amino acid metabolism disorder" describe how the disorder was evaluated.   2.2. Interventions: Make clear the selection criteria for the 6 non-essential amino acids. Why the proportions of amino acids were different for each. Why was alanine in greater amounts and serine and glycine in less whether there are reports that glycine could improve the synthesis of type II collagen?   As I understand it, the patients themselves who filled out the questionnaires? This would not be the best way to obtain the data, as there is a bias when comparing the questions being asked by an interviewer.   2.5. Statistical analysis. Why didn't you compare week 4 to week 12? This analysis would make sense, as it would show whether intervention after 4 weeks would be necessary or not.   It would be interesting to make a comparison between different age groups, since over the years it is known that joint wear increases.   I believe that the chosen term "healthy individuals" would not be the best way to describe the participants. What sense does the study make if the individuals are healthy? If that were the case, I believe it would make more sense to understand the mechanisms by which non-essential amino acids could assist in collagen formation, for example. But as the focus is on the role of these amino acids in improving pain, discomfort and stiffness, mainly, even if the participants have not been diagnosed with a joint disease, they do have some symptoms.   As the authors highlighted, the study has several limitations. In my view, among them, the lack of data on diet is the most serious, as this is a major bias and may be bringing all the differences found between the groups. Wouldn't it be possible to get this data?

Author Response

Thank you for your careful review. Below please find a detailed response to each comment, indicated by blue. The manuscript has been revised, so please see the attachment.

Comment 1: 2.1. Study design and participants. In the exclusion criterion "(10) had an amino acid metabolism disorder" describe how the disorder was evaluated.

Response 1: Participants self-reported the presence of congenitally abnormal amino acid metabolic disorders such as phenylketonuria. A sentence was added to the manuscript to clarify. (line 82 on page 2)

Comment 2: 2.2. Interventions: Make clear the selection criteria for the 6 non-essential amino acids. Why the proportions of amino acids were different for each. Why was alanine in greater amounts and serine and glycine in less whether there are reports that glycine could improve the synthesis of type II collagen?

Response 2: The proportions of 6 non-essential amino acids were nearly identical, with the exception of alanine. In the non-clinical study, we only confirmed the effects of the L form of alanine; however, only the DL form of alanine is used as a food additive in Japan. Therefore, the proportion of DL-alanine in 6AA was approximately twice that of other non-essential amino acids. Thus, the final 6AA proportion was decided regarding cost, taste and safety.

Comment 3: As I understand it, the patients themselves who filled out the questionnaires? This would not be the best way to obtain the data, as there is a bias when comparing the questions being asked by an interviewer.

Response 3: The participants filled out Visual Analogue Scale (VAS) and the Japanese Knee Osteoarthritis Measure (JKOM), both of which are well-established self-report measures utilized to assess knee joints. Our reasons for selecting VAS and JKOM are mentioned on lines 123 and 140 on page 3. Doctors evaluated the Japanese Orthopaedic Association (JOA) scores of the same participants over the experimental period to minimize bias.

Comment 4: 2.5. Statistical analysis. Why didn't you compare week 4 to week 12? This analysis would make sense, as it would show whether intervention after 4 weeks would be necessary or not.

Response 4: Thank you for your critique. We observed that continuous intake of 6AA improved at both W4 and W12. We used Student’s t-test to assess between-group differences from W0 at W4 and W12 weeks. We found that a 4-week 6AA supplementation period was sufficient to achieve the desired effect. In addition, the LMM analysis indicated that the 6AA effects were significant at all the primary endpoints; thus, it appears that longer intake duration corresponds to greater improvements over the 12-week intervention. Therefore, supplementation durations longer than 4 weeks might be more effective.

Comment 5: It would be interesting to make a comparison between different age groups, since over the years it is known that joint wear increases.

Response 5: Thank you for your helpful comment. We recognize the value of age groups comparisons for confirming potential differences in the effects of 6AA relative to age. The current study is insufficiently powered for comparison of individual age groups. Also, the primary aim of this study was to clarify the effects of 6AA across adult participants of varying ages (20–64 years). Our statistical analysis plan was developed specifically with this objective in mind. Future investigations are needed to clarify the effects of 6AA in different age groups.

Comment 6: I believe that the chosen term "healthy individuals" would not be the best way to describe the participants. What sense does the study make if the individuals are healthy? If that were the case, I believe it would make more sense to understand the mechanisms by which non-essential amino acids could assist in collagen formation, for example. But as the focus is on the role of these amino acids in improving pain, discomfort and stiffness, mainly, even if the participants have not been diagnosed with a joint disease, they do have some symptoms.

Response 6: Thank you for your important comment. We agree that the term ‘healthy individuals’ could be viewed inappropriately; however, we did not include methods necessary to objectively determine if the participants were completely healthy. Thus, we changed the term ‘healthy individuals’ to ‘adults’ or ‘otherwise healthy individuals experiencing joint discomfort’ in the title and throughout the main body of the manuscript.

Comment 7: As the authors highlighted, the study has several limitations. In my view, among them, the lack of data on diet is the most serious, as this is a major bias and may be bringing all the differences found between the groups. Wouldn't it be possible to get this data?

Response 7: This is a crucial point. Unfortunately, we do not have access to the dietary data the reviewer requests. However, dietary data were obtained in another interventional study of 6AA completed by this group. In that study, 6AA improved joint condition regardless of diet (data to be published shortly). Thus, we doubt that differences in dietary intake would affect the results of the present study.

Reviewer 2 Report

no coment

Author Response

Thank you for reviewing our manuscript.

Reviewer 3 Report

Dear authors,

First, I would like to express sincere gratitude to get an opportunity to review your manuscript.

The effort of the author is appreciated. It was my pleasure to assess your manuscript. Congratulations on the selection of the subject of your manuscript, and also on performing a RCT. After assessing the manuscript, the following issue raised my concerns or represent suggestions that in my opinion could increase the quality of the manuscript:

-       In the abstract section of the manuscript, you mention “20 and 60 years” and the in the study design and participants “20–64 years”. It is possible that I had misunderstood.

Author Response

Thank you for your careful review. We have made this correction. Please see the attachment.

Round 2

Reviewer 1 Report

The issues raised were clarified. I believe the manuscript is ready for publication.